# Protocol for a randomised controlled trial of Subacromial spacer for Tears Affecting Rotator cuff Tendons: a Randomised, Efficient, Adaptive Clinical Trial in Surgery (START:REACTS)

Andrew Metcalfe [1,2] Elke Gemperle Mannion,[1] Helen Parsons [1] Jaclyn Brown,[1] Nicholas Parsons [1] Josephine Fox,[3] Rebecca Kearney [1,2] Tom Lawrence,[2] Howard Bush,[2] Kerri McGowan,[2] Iftekhar Khan [1] James Mason [1] Charles Hutchinson,[1] Simon Gates,[1,4] Nigel Stallard,[1] Martin Underwood [1] Stephen Drew[2]

For numbered affiliations see end of article.

**Correspondence to**
Mr Andrew Metcalfe;
A.Metcalfe@warwick.ac.uk

## ABSTRACT

**Introduction** Shoulder pain due to irreparable rotator cuff tears can cause substantial disability, but treatment options are limited. A balloon spacer is a relatively simple addition to a standard arthroscopic debridement procedure, but it is costly and there is no current randomised trial evidence to support its use. This trial will evaluate the clinical and cost-effectiveness of a subacromial balloon spacer for individuals undergoing arthroscopic debridement for irreparable rotator cuff tears. New surgical procedures can provide substantial benefit to patients. Good quality randomised controlled trials (RCTs) are needed, but trials in surgery are typically long and expensive, exposing patients to risk and the healthcare system to substantial costs. One way to improve the efficiency of trials is with an adaptive sample size. Such methods are well established in drug trials but have rarely, if ever, been used in surgical trials.

**Methods and analysis** Subacromial spacer for Tears Affecting Rotator cuff Tendons: a Randomised, Efficient, Adaptive Clinical Trial in Surgery (START:REACTS) is a participant and assessor blinded, adaptive, multicentre RCT comparing arthroscopic debridement with the InSpace balloon (Stryker, USA) to arthroscopic debridement alone for people with a symptomatic irreparable rotator cuff tear. It uses a group sequential adaptive design where interim analyses are performed using all of the 3, 6 and 12-month data that are available at each time point. A maximum of 221 participants will be randomised (1:1 ratio), this will provide 90% power (at the 5% level) for a 6 point difference in the primary outcome; the Oxford Shoulder Score at 12 months. A substudy will use deltoid-active MRI scans in 56 participants to assess the function of the balloon. Analysis will be on an intention-to-treat basis and reported according to principles established in the Consolidated Standards of Reporting Trials statement.

**Ethics and dissemination** NRES number 18/WM/0025. The results will be disseminated via peer-reviewed

### Strengths and limitations of this study

► Multicentre randomised trial of a subacromial spacer balloon following debridement, compared with debridement alone, for irreparable rotator cuff tears of the shoulder.
► Participant-assessor blinding, including blinded operation notes with novel unblinding mechanism.
► Mechanistic MRI substudy of 56 participants with images at 8 weeks and 6 months after surgery.
► Statistical adaptive design, with hard stopping rules for futility or efficacy based on emerging outcomes.

publications, presentations at conferences, lay summaries and social media.

**Trial registration number** ISRCTN17825590

## INTRODUCTION

### Subacromial spacer balloons

Shoulder pain is a common and disabling problem. The UK population prevalence of shoulder pain is approximately 16%.[1] Rotator cuff disease accounts for 70%–85% of this.[2–4] People with a symptomatic rotator cuff tear typically have pain, restricted movement, loss of strength and disability. The condition is associated with substantial expense to society through both costs of treatment and loss of work (both paid and unpaid).[5–8] Rotator cuff repair is a widely accepted treatment for symptomatic rotator cuff tears.[9 10] Some tears cannot be surgically repaired, these are called irreparable tears.

Symptomatic irreparable rotator cuff tears are a challenging condition to treat. Treatment options including physiotherapy, injections, arthroscopic debridement, partial repair, muscle transfers, interposition grafts and even shoulder replacements.[11–14] Arthroscopic debridement is commonly used and case series suggest it may be beneficial, but it remains a controversial option, with little randomised controlled trial (RCT) data on its use in the irreparable tear population.[15–17]

In 2013, the InSpace subacromial balloon spacer (Stryker, Michigan, USA) was introduced into UK orthopaedic practice as a potential treatment option for people with irreparable tears of the rotator cuff. In May 2016, an interventional procedure guidance document was published by the UK National Institute for Health and Care Excellence (NICE), which found that there was very limited evidence for its use. Therefore, the device was limited to research use only and a recommendation was made to assess its effectiveness.[18] It has not yet received US Food and Drug Administration clearance in the USA, but it is a widely used treatment elsewhere in the world.

The InSpace device is a saline-filled balloon made of biodegradable (dissolvable) synthetic material. It is inserted above the main joint of the shoulder at the end of an arthroscopic debridement after an irreparable tear has been identified. It is simple to deploy and adds relatively little time to the operation.[19 20] It cushions the humerus from pressing on the bone above it (the acromion) when the deltoid is active and during abduction of the arm, potentially reducing pain. It may also assist in the biomechanics of the shoulder, resisting proximal migration of the humerus under deltoid activity. The device begins to degrade and deflate from 3 months after surgery. During this time, it is thought to improve rehabilitation of the remaining rotator cuff and deltoid, so that when the device deflates, the biomechanics of the shoulder are better preserved.

The safety of the device in rodents has been established, with only one adverse event (AE): a fibrosarcoma that was thought to be unique to rodents.[21] Proof of concept has been established in a series of 24 irreparable cuff tear cases in Slovakia in 2012, with 5-year follow-up results published in 2016.[22 23] The device has been used in a number of centres across the UK, with multiple conference abstracts from the UK.[24–26] These have demonstrated improvements in outcomes from baseline. Complications such as balloon displacement and non-cyst forming synovitis have been reported in a small number of cases (3 out of 61). One retrospective non-randomised study of 23 patients (12 with the balloon) showed an improvement in outcomes compared with debridement alone.[27] There are no completed RCTs.

The benefits found in case series may not be unique to the InSpace balloon, although the relative effectiveness of the balloon in comparison to non-operative care or acromioplasty is not known and it could give substantial improvements.[17]

There is one other RCT in progress for this device—a company funded study in the USA which will recruit 184 participants, comparing partial cuff repair with balloon as a stand-alone intervention. Results are expected in mid-2020. Partial cuff repair is not a technique that is widely used in the UK and is not an appropriate comparator in a UK context.

## Trial designs in new surgical procedures

The safe introduction of new surgical procedures are essential to the delivery of high quality surgical care for patients. New procedures can result in a step-change improvement in treatment, but also introduce new risks and substantial costs. Major harm can be done when a misunderstood or well-intentioned intervention is used widely across the health service before it is formally evaluated.[28 29]

While pharmaceuticals must undergo rigorous clinical trials before introduction, this is not the case for surgical procedures.[30] When surgical procedures are assessed rigorously, studies tend to be large multicentre, randomised trials which typically need to recruit over extended time periods.[29 31] Improvements in the efficiency of delivering trials of surgical interventions would provide earlier answers to these important clinical questions, providing benefits to patients and making better use of healthcare resources.

The use of adaptive trial designs has been encouraged by major journals, the US Food and Drug Administration, and National Institute of Health Research panels.[32–34] Adaptive trial designs allow prospectively planned modifications, such as stopping the study early or dropping an intervention, based on emerging findings as the trial proceeds, while preserving the scientific validity and integrity of the trial. This flexible strategy typically reduces costs and shortens time scales, without compromising the integrity, statistical power or rigour of the study.[32 35–37]

In this trial, we are using a novel statistical adaptive design approach for the assessment of a new surgical procedure. The statistical principles are laid out in a 2019 methodology paper.[38] Briefly, the design uses early (3 and 6-month Oxford Shoulder Score, OSS) endpoint data to support emerging outcome data for the primary endpoint (12-month OSS) to make decisions about stopping for either futility or efficacy at preplanned interim analyses that occur before the end of recruitment. In the current study, we will apply these design principles to a multi-centre clinical trial of a new surgical procedure.

## Aim

To assess the clinical effectiveness, cost-effectiveness and safety of a subacromial spacer balloon for patients with symptomatic irreparable tears of the rotator cuff.[18]

## METHODS

### Trial design

The Subacromial spacer for Tears Affecting Rotator cuff Tendons (START) study is a participant and assessor blinded, adaptive, multicentre RCT based in the UK comparing arthroscopic debridement using the InSpace balloon to arthroscopic debridement alone, performed using the Randomised, Efficient, Adaptive Clinical Trial in Surgery (REACTS) framework.

### Patient and public involvement

Patient involvement has been a core part of the design and delivery of the study, and will remain so, including in the interpretation and dissemination of results. In the planning stages, we engaged with multiple patients who had previously undergone rotator cuff surgery and their insights helped establish the design of the study, especially the outcomes. We reviewed patient facing materials with many shoulder patients before they were finalised. One of the coauthors is a patient and represents the patient view in trial management meetings, two patients sit on our steering committee. We will produce patient and public-focused summaries of the research and disseminate this widely.

### Objectives

#### Clinical objectives

▶ Our primary clinical objective is to quantify and draw inferences on observed differences between arthroscopic debridement of the subacromial space and arthroscopic debridement with insertion of the InSpace balloon (Stryker, Michigan, USA) 12 months after surgery, using the OSS as the primary outcome measure.[39 40]

  Secondary clinical objectives

▶ To quantify and draw inferences on observed differences between arthroscopic debridement and arthroscopic debridement with insertion of the InSpace balloon (Stryker, Michigan, USA) on: shoulder function; patient-reported outcome measures; AEs and resource use at 3, 6 and 12 months.
▶ To perform an economic analysis, assessing the comparative cost-effectiveness of the two treatments.
▶ To compare the acromio-humeral distance on MRI scans in a sample of participants with and without the balloon at 8 weeks and 6 months after treatment, to assess the proposed mechanism of action of the balloon when it is still inflated (at 8 weeks) and to determine if the effect persists when it has deflated (at 6 months).

#### Methodological objectives

▶ The primary methodological objective is to develop and implement appropriate statistical tools to allow an efficient adaptive clinical trial design, with the potential for early stopping for either futility or efficacy, using OSS data available at 3, 6 and 12 months.[38]

### Ethics, registration and oversight

The trial will conform to the principles of the Declaration of Helsinki and to MRC Good Clinical Practice guidelines. It will also comply with all applicable UK legislation and University of Warwick Standard Operating Procedures (SOPs). Trial oversight is provided by a data monitoring committee (DMC) and trial steering committee (TSC), both are made up of a majority of independent members and are conducted according to Warwick SOPs. Monitoring and audit will be undertaken by the sponsors according to a monitoring plan. The trial received full research ethical approval (RES number 18/WM/0025) on 13 February 2018, prior to commencing recruitment which is ongoing. Amendments will be communicated to sites by the coordinating team.

### Outcome measures

#### Primary outcome

The OSS at 12 months. The original study design was based around the Constant score, however the trial management group decided in March 2020, in light of the coronavirus outbreak, to revise this to the OSS. As the Constant score is a face-to-face measure usually taken in hospital clinics, it would have exposed participants to unnecessary risk during the height of the pandemic. The decision was agreed by both the TSCs and DMCs prior to the change.

  The OSS has been well validated and used in high-impact randomised trials previously, it correlates well with the Constant score, both are similarly responsive and have comparable effect sizes in rotator cuff pathology.[39–43] Based on our meta-analysis of outcomes for randomised trials, shoulder scores typically reach a plateau at 12 months after any intervention for a rotator cuff tear.[44] The 24-month scores do not give sufficient additional value to justify the increase in costs and delay in the trial result that would be required had 24 months been used as the primary outcome.

#### Secondary outcomes

▶ The OSS at baseline, 3, 6 and 24 months.
▶ The Constant Score at baseline, 3, 6 and 12 months. A standardised protocol for the objective component of the score has been developed based on the work of Moeller *et al* with training provided for all sites.[45–47]
▶ Range of pain-free movement of the shoulder at baseline, 3, 6 and 12 months measured using a long-arm goniometer (12 ½").
▶ Strength of shoulder abduction and flexion at baseline, 3, 6 and 12 months measured using a supplied IsoForceControl EVO$_2$ dynamometer (Herkules Kunstoff, Switzerland).
▶ Western Ontario Rotator Cuff (WORC) index at baseline, 3, 6, 12 and 24 months.[48]
▶ Health utility assessed using 5-level EuroQol 5 dimension score (EQ-5D-5L) at baseline, 3, 6, 12 and 24 months.[49 50]
▶ Healthcare resource collected at 3, 6 and 12 months.

► Patient Global Impression of Change (PGIC) score, collected at 3, 6, 12 and 24 months. A simple 7-point scale assessing perception of improvement.[51]
► Analgesia use (drug and approximate frequency) collected at baseline and 3, 6 and 12 months.
► MRI Scans (substudy of 56 patients, 6 weeks and 6 months postsurgery): see the 'MRI substudy' section.
► AEs will be collected from site reports as they occur throughout the first 12 months, and from participants in the 3, 6 and 12 months questionnaires.

Patient-reported outcome measures (OSS, WORC, EQ-5D-5L, PGIC) will be collected at 24 months as a secondary outcome, these will be published separately and will not delay publication of the primary 12-month outcome data. A detailed data management plan has been prepared following Warwick SOPs and will be available on request.

## Eligibility criteria
### Inclusion criteria
1. Rotator cuff tear deemed by the treating clinician to be technically irreparable*. Many factors other than size influence whether a tear can be repaired (such as chronicity, retraction of the tendon ends, fat infiltration in muscle). However, a potential participant who has a tear that is technically repairable, such as a small tear, but is unsuitable for repair due to age or comorbidities, is not eligible for this study.
2. Intrusive symptoms (pain and loss of function) which in the opinion of the treating clinician warrants surgery.
3. Non-operative management has been unsuccessful. The exact nature of non-operative management will be left for the treating clinician to decide.

### Exclusion criteria
4. Advanced gleno-humeral osteoarthritis (OA) on preoperative imaging (in the opinion of the treating clinician). Advanced glenohumeral OA may be interpreted as Kellgren Lawrence grade 3 or 4 changes on routine preoperative radiographs,[52] or the MRI equivalent if radiographs have not been taken.
5. Subscapularis deficiency* defined as a tear involving more than the superior 1 cm (approximately) of the subscapularis if repaired, or any tear that is not repaired.
6. The treating clinician determines that interposition grafting or tendon transfers are indicated.
7. Pseudoparalysis (an inability to actively abduct or forward flex up to 20°), as determined by the treating clinician.
8. Unrelated, symptomatic ipsilateral shoulder disorder that would interfere with strength measurement or ability to perform rehabilitation.
9. Other neurological or muscular condition that would interfere with strength measurement or ability to perform rehabilitation, in the opinion of the treating clinician.

10. Previous proximal humerus fracture that could influence shoulder function, as determined by the treating clinician.
11. Previous entry into the present trial (ie, other shoulder).
12. Unable to complete trial procedures.
13. Age under 18.
14. Unable to consent to the trial.
15. Unfit for surgery as defined by the treating clinician.

*criteria regarding whether the tear is technically repairable, and the integrity of the subscapularis, are unreliably assessed by preoperative imaging and will be reassessed in theatre prior to randomisation. People not eligible to be enrolled in the trial will be treated according to the judgement of the surgeon at the time.

## Participant identification, screening and withdrawals
Potential participants will be identified by the attending clinical team by clinicians in intermediate or secondary care clinics, or from the surgical waiting list. The attending clinician will confirm appropriateness for study eligibility based on clinical assessment and standard care preoperative imaging for that site.

All potential participants who meet the study entry criteria will be checked for eligibility and recorded on the monthly screening log. Potential participants who are willing to be approached by a suitably trained member of the research team will be provided with verbal and written information about the study, and will have the opportunity to discuss and ask questions about the study prior to informed consent being obtained.

Eligibility for the study is confirmed by the operating surgeon intraoperatively and patients may be excluded at this stage if the surgeon finds that the rotator cuff tear can be repaired. These excluded participants will be informed by letter that they are no longer taking part in the study. Baseline data will be retained to explore any differences between those who were deemed ineligible at surgery to those who were randomised.

Participants randomised into the study will be allowed to withdraw from follow-up at any time, without prejudice. This will have no effect on their current or future care.

## Randomisation
Participants are randomly allocated on a 1:1 basis to the two treatment groups via a central computer-based randomisation system provided by Warwick Clinical Trials Unit (WCTU) independent of the study team. A minimisation algorithm is used to determine participant allocation, using site, gender, age group (<70 years and ≥70 years) and cuff tear size (as assessed by the operating surgeon, ≥3 cm or <3 cm) as factors, with a random element included to provide a 70% chance that the participant will receive the treatment that minimises the imbalance, to ensure unpredictability.

Randomisation will be performed by theatre staff, after the intraoperative findings have been checked (including cuff tear size) and eligibility is confirmed. Staff use an

online system in a separate room to maintain blinding, with a 24-hour back-up automated telephone system available.

## Trial treatment(s)/intervention
### Group 1: Standard arthroscopic debridement (control)
The control group will be an arthroscopic debridement of the subacromial space with removal of inflamed tissue (bursectomy) and unstable remnants of the torn tendon, limited bone resection of the acromion, retention of the coracoacromial ligament and biceps tenotomy (if not already torn). The anaesthetic will be left to the choice of the anaesthetist; this may include general or local anaesthesia. Surgeons may use their normal surgical technique, within the confines described in a trial specific surgical guideline (available at the trial website or at request from Warwick CTU).

### Group 2: Standard arthroscopic debridement plus insertion of InSpace Balloon (intervention)
Arthroscopic debridement, as described above, with insertion of the InSpace Balloon performed by subspecialty trained shoulder surgeons. The same arthroscopic debridement will be performed as described for the control group. If allocated to the balloon procedure, the company's recommended surgical technique will be followed for sizing, insertion and deployment of the balloon, as documented in the surgical manual.

For both groups, fidelity will be assessed with an operative record form and arthroscopic photographs taken at the end of debridement and just before balloon inflation, and the number of physiotherapy visits for each participant will be documented in both arms.

## Rehabilitation
Postoperative rehabilitation for both groups will be blind to treatment allocation and will include standardised postoperative information, home exercises and the offer of a minimum of three face-to-face physiotherapy appointments. Additional physiotherapy will be at the discretion of the trial sites. A physiotherapy trial manual will be followed to standardise rehabilitation progression. All materials were developed through a process of collecting current National Health Service (NHS) protocols, manufacturer protocols, scoping the literature and expert consensus.

## Blinding
Treatment allocation will be blinded for both patients and assessors, only the surgical teams at the time of the operation will be aware of the allocation. Theatre staff are asked not to discuss the balloon and to communicate the allocation by using methods such as holding up a piece of paper on which the allocation is clearly written. If participants are awake, drapes are used to obscure the participant's view and arthroscopic screens will be positioned in such a way that the patient is unable to see the procedure.

The incisions required for the two operations are similar and there is no external way in which the patients

will be able to detect the presence or absence of the balloon, except for the size of the lateral portal. One of the incisions (the lateral portal) will need to be slightly larger to insert the balloon—1.5 cm as opposed to 1 cm. A 1.5 cm incision will be used for all participants, which is a very small change from standard care and is very unlikely to have a negative effect on any participant. Incisions will therefore be the same for both groups.

The operation note will be blinded to prevent accidental unblinding of the patient (eg, in the discharge information or during postoperative physiotherapy). A standard recommended operation note template will be given to all sites adjusted to fit their local operation note systems. The details of the operation related to the balloon will be recorded in a secure online form easily accessible to the surgeon.

Unblinding may very rarely be required in an emergency situation, such as an overnight admission for suspected postoperative infection. Unblinding will be performed only by NHS staff in an emergency situation, by using a predefined web-based system, from a link inserted in the operation note. A two-way secure verification process will be performed using email, and an access code will be emailed only to an active NHS email address. A full explanation of the clinical circumstances and the need for access to data will be requested by the trial team for audit and monitoring purposes from the person who performed the unblinding, and the principal investigator for the site will be informed. The system has been designed and tested by the WCTU programming team to ensure that it is both secure and fully functional.

Participants will be asked at the 12-month time point, after collection of the primary outcome, if they were aware of their allocation.

## End of trial
The trial will end when all participants have completed their 24-month follow-up, but the results will be published after the 12-month outcomes have been analysed, with a secondary report after collection of 24-month outcomes.

The trial may be stopped prematurely if: (1) mandated by the Ethics Committee; (2) there is an unexpected major safety concern; (3) following recommendations from the DMC or TSC, which may include meeting the adaptive design boundaries or (4) funding for the trial ceases

Once a decision to stop has been taken, those within a short waiting time to surgery may be randomised at the discretion of the DMC and TSC, but after that participants will be treated according to the judgement of their clinician.

## MRI substudy
Participants selected for inclusion in the MRI substudy will undergo two research MRI scans at 8 weeks and 6 months post-operatively. MRI scans were preferred to X-rays for this purpose as measures taken from X-rays would be prone to error, primarily due to variation in

the angle between the shoulder joint and the beam of the X-ray. This will assess the mechanism of action of the balloon, when the balloons are likely to be still inflated (but when acute postoperative pain has subsided), and when they are likely to have fully deflated, to see if the proposed mechanism for ongoing improvement is maintained. Passive imaging alone will not be adequate to demonstrate the function of the balloon and imaging will also need to be performed when the deltoid muscle is active, producing a proximally directed force on the humerus.[53 54]

For this study, we have developed (and piloted) a novel dynamic approach to assessing the function of the rotator cuff using MRI imaging under a mild deltoid load, to specifically assess the mechanism of the balloon. A separate paper on the pilot study for the trial is in preparation and will describe these methods in more detail.

All participants will be eligible (unless they have a contraindication to MRI, or do not want to take part in the substudy), but the substudy will only be undertaken at a proportion of participating sites. The primary outcome will be the minimum acromiohumeral distance on the 'deltoid-active' coronal sequences at 6 months, a reliable and proven measure.[55–57] The balloon is expected to have deflated by 6 months.[58] Secondary measures will be acromiohumeral distance on passive images, and the change in acromio-humeral distance between active and passive images. The position of the balloon will be assessed to check for migration and consistency of placement relative to the acromion.

### Adverse events, adverse device effects, serious adverse device effects

Adverse events (AEs), serious AEs (SAEs), adverse device effects (ADE) and serious ADEs (SADE) will be defined using standard accepted criteria. An unanticipated SADE (USADE) is defined as an SADE which by its nature, incidence, severity or outcome has not been identified in the current version of the risk analysis report.

For the purposes of this trial, AEs will be recorded for any participant where it is thought there may be a relationship between the trial interventions or the condition being studied (in this case, any shoulder condition or related to the anaesthetic). These include device specific complications such as balloon migration, which will be recorded if it is identified by clinical teams from their normal practice.

All SAEs, SADEs and USADEs occurring from the time of randomisation until 12 months postrandomisation must be communicated to the sponsor within 24 hours of the research staff becoming aware of the event. Events will be followed up until the event has resolved or a final outcome has been reached.

Where participants have been lost to follow-up at or beyond the 12-month time point, and data on AEs can therefore not be recorded, the participants general practitioner (GP) will be contacted and a short form requesting any information or health record that could be an AE will be requested from the GP, as well as confirmation of the current contact details of the participant.

## STATISTICAL ANALYSIS
### Power and sample size

The initial sample size calculations were based on the Constant Score. The target difference that was chosen for the Constant Score was 10 units, as has been widely used for other trials, and the SD was taken to be 20, giving a moderate standardised mean difference of 0.5.[17 23 59 60] Anchor-based studies have estimated the target difference for the OSS as 6 and an SD of 12 has been observed in multiple studies, therefore, a moderate standardised mean difference of 0.5 remains appropriate.[41–43 60]

For a costly invasive procedure of this nature, a smaller standardised mean difference is unlikely to be considered worthwhile. For a power of 90% and a (two-sided) type I error rate of 5% a study with a conventional fixed design (ie, with no possibility of stopping early), assuming an approximate normal distribution for the score data, would require 170 participants.

In order to assess the design characteristics and estimate the required sample size for the planned adaptive design, we undertook a large simulation study (see Parsons et al[38]). As we expect correlations between time points (based on Karthikeyan et al) and effect sizes to be equivalent between the Constant Score and Oxford Scores, these simulations remain valid, despite the change of primary outcome score.[41 60] The simulations showed that an adaptive design that allowed the possibility of early stopping for efficacy and/or futility, was feasible for the START:REACTS study.

Based on an assumed modest correlation between 3-month, 6-month and 12-month OSSs equal to 0.5, and SD of 12 for both three and 6-month scores, the simulations for the selected adaptive design indicated that 188 participants would be sufficient to detect a 6 point difference in OSS between treatment arms with 90% power, and 5% (two-sided) type I error rate. Allowing for 15% lost to follow-up, while striving to keep this below 10%, gives a maximum study sample size of 221.

The sample size for the MRI substudy is based on the study by Gumina et al[55], where the minimum acromio-humeral distance had an SD of 1.72 mm. To observe a minimum important difference of 1.5 mm (larger than the minimum detectable change of 1.3 mm established elsewhere[61]) with an alpha of 0.05 at 80% power, assuming a lost to follow-up at 6 months of 20%, requires 56 participants for the MRI substudy.

### Statistical analysis plan

All data will be analysed and reported in accordance with the Consolidated Standards of Reporting Trials (CONSORT) guidelines.[62 63] A detailed statistical analysis plan and a data sharing plan will be agreed with the DMC prior to any formal analyses being conducted.

Baseline data will be summarised to check comparability between treatment arms. Standard statistical summaries (eg, means and SD, dependent on data type) will be presented for the primary outcome measure and all secondary outcome measures.

The primary analysis will investigate differences in the OSS 12 months after surgery between the two treatment groups. It will be conducted on an intention-to-treat basis following the methods, test statistics and boundaries described by Parsons *et al*.[38] To preserve the integrity of the study, the exact boundaries used for testing are given in the adaptive charter known only to the study team and independent DMC. Briefly, if the study recruits to target, the method set out by Parsons *et al*[38] will be used to calculate the boundaries and if the study stops early, testing will proceed using boundaries calculated by the deletion method for overrunning analysis, described by Whitehead, with clinical inferences following directly from widely used methods for unbiased estimates and confidence intervals in group sequential trials.[64–66]

The primary analysis will be augmented by calculating adjusted estimates of treatment group differences (with 95% CIs) for the OSS using a mixed-effects model. The mixed-effects model will include a random effect for the recruiting centre, and fixed effects for the variables of interest included in the minimisation algorithm, patient age, gender and size of tear. In addition, subject to the observed data, sensitivity analyses such as per-protocol analyses or multiple imputation will also be undertaken. If undertaken, all multiple imputation assumptions will be justified and reported. Estimates of efficacy for the other outcome measures will follow this approach to analysis.

Prespecified subgroup analyses will be undertaken to assess whether there is evidence that the intervention effect differs with respect to:

► The size of the rotator cuff tear as measured at the start of surgery, defined as large or massive cuff tear (≥3 cm) or moderate to small (<3 cm).
► Gender.
► Age (>70 or <70).

These have been chosen as they are either important to the function of the intervention (cuff tear size) or to interpretation (gender and age). The subgroup analyses will follow the methods described for the mixed effects model of the OSS above, with additional interaction terms incorporated into the mixed-effects regression model to assess the level of support for these hypotheses. The study is not powered to formally test these hypotheses, so they will be reported as exploratory analyses only, and as subsidiary to the analysis reporting the main effects of the intervention in the full study population.[67]

Interim analyses will only be undertaken following the principles laid out by Parsons *et al*.[38] Details of the timings or settings for the interim analyses will be treated as confidential information, so as not to prejudice the outcome of the trial based on the decision to stop the study or proceed, but will be recorded in the DMC minutes and on date-stamped internal documents. If a decision to stop the study early is made (for reasons other than safety), it will not be communicated outside of the DMC closed meeting whether this is for efficacy or futility.

## HEALTH ECONOMIC EVALUATION

The economic component will include a standard health policy-relevant economic analysis and an exploration of how early economic data can be used to support decision making through the use of an adaptive design.

A prospective economic evaluation will be integrated into the trial and adhere to the recommendations of the NICE Reference Case.[68] Mechanisms of missingness of data will be explored and multiple imputation methods will be applied to impute missing data. Imputation sets will be used in bivariate analysis of costs and quality-adjusted life years (QALYs) to generate incremental cost per QALY estimates and confidence regions.[69–72] It is anticipated that incremental costs and benefits will be captured within the trial and that extrapolated modelling will not be required.

Relatively little research has been conducted on how economic data available at interim time points can inform interim decision making. Currently, methods used are based mainly on net monetary benefit approaches using value of information methods.[73 74] We will compare various methods using economic data (costs and QALYs) and clinical data to evaluate the practical implications and operating characteristics of stopping a trial early based on cost-effectiveness data alone, efficacy data alone or a combination of cost-effectiveness and efficacy. We will use the trial to evaluate putative analytical methods, as set out within a prospectively written health economic analysis plan and carry out parallel interim analyses, separate from the real trial analyses, exploring how interim decisions might have been influenced. As such health economic decision rules for adaptive designs are less widely understood and used in the literature, the properties of these methods will be developed further for future trials.

## DISSEMINATION AND PUBLICATION

The trial will be reported in accordance with the CONSORT guidelines and the 2018 CONSORT extension for adaptive trials.[62 63]

The results will be submitted to a high impact peer-reviewed journal with authorship following International Committee of Medical Journal Editors (ICMJE) recommendations, and will be presented at national and international meetings such as the British Elbow and Shoulder Society, the British Orthopaedic Association and the American Academy of Orthopaedic Surgeons. The investigators will share data (with associated coding library) used in developing the results on request to the chief investigator, subject to a formal data sharing agreement being in place. We will only share anonymised record

level data with those who have received ethical clearance from their host institution.

To inform patients and the public, we intend to produce a lay summary, which will be made available in the trial hospitals and to patients involved in the trial. In addition, we will publicise the work through social media outlets (eg, Facebook and twitter) as well as websites such as Patient.co.uk.

**Author affiliations**
¹Warwick Medical School, University of Warwick, Coventry, UK
²University Hospitals Coventry and Warwickshire NHS Trust, Coventry, UK
³Patient Representative, Durham, UK
⁴Cancer Research UK Clinical Trials Unit, University of Birmingham, Birmingham, UK

**Contributors** All the authors helped to developed the trial protocol and all have contributed to the writing of the manuscript and approved its final version. TL and SD first identified the clinical question. The clinical question and adaptive design components were developed by AM, TL, SD, MU, HP, NP, NS, RSK, JM, IK and SG in close collaboration with JF throughout. These same coauthors developed the study methodology with EGM, JB, KM, HB and CEH. CEH leads the MRI substudy with AM. This protocol was written following the SPIRIT protocol guidance. AM is the chief investigator and main grant holder for this study.

**Funding** This project (project reference 16.61.18) is funded by the Efficacy and Mechanism Evaluation (EME) Programme, an MRC and NIHR partnership. The trial is cosponsored by the University of Warwick and University Hospitals Coventry and Warwickshire NHS trust.

**Disclaimer** The views expressed in this publication are those of the author(s) and not necessarily those of the MRC, NIHR or the Department of Health and Social Care.

**Competing interests** Stryker is providing 50 free InSpace balloons for the study, to offset some of the costs of delivery of the study in participating hospitals. Contracts are in place to ensure the full independence of the trial team with regard to study design and delivery, analysis and reporting of results, aligning to the NIHR standard agreement. The chief investigator and multiple coauthors (HP, EGM, JB, JF, JM, CEH and MU) are applicants on another NIHR grant with similar contractual relationships with Stryker to support treatment costs but not research costs, in which the full independence of the trial team is protected by mutually agreed contracts. MU was Chair of the NICE accreditation advisory committee until March 2017 for which he received a fee. He is chief investigator or coinvestigator on multiple previous and current research grants from the National Institute for Health Research. He is an NIHR Senior Investigator. He is an editor of the NIHR journal series, and a member of the NIHR Journal Editors Group, for which he receives a fee.

**Patient and public involvement** Patients and/or the public were involved in the design, or conduct, or reporting, or dissemination plans of this research. Refer to the Methods section for further details.

**Patient consent for publication** Not required.

**Provenance and peer review** Not commissioned; externally peer reviewed.

**ORCID iDs**
Andrew Metcalfe http://orcid.org/0000-0002-4515-8202
Helen Parsons http://orcid.org/0000-0002-2765-3728
Nicholas Parsons http://orcid.org/0000-0001-9975-888X
Rebecca Kearney http://orcid.org/0000-0002-8010-164X
Iftekhar Khan http://orcid.org/0000-0001-6041-8837
James Mason http://orcid.org/0000-0001-9210-4082
Martin Underwood http://orcid.org/0000-0002-0309-1708

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
