## [Reviewer comments · BMJ Open]

ARTICLE DETAILS

TITLE (PROVISIONAL)	Protocol for a randomised controlled trial of Subacromial spacers for Tears Affecting Rotator cuff Tendons: a Randomised, Efficient, Adaptive Clinical Trial in Surgery (START:REACTS)
AUTHORS	Metcalfe, Andrew; Gemperle Mannion, Elke; Parsons, Helen; Brown, Jaclyn; Parsons, Nicholas; Fox, Jo; Kearney, Rebecca Samantha; Lawrence, Tom; Bush, Howard; McGowan, Kerri; Khan, Iftekhar; Mason, James; Hutchinson, Charles E; Gates, Simon; Stallard, Nigel; Underwood, Martin; Drew, Stephen

VERSION 1 – REVIEW

REVIEWER	Xinning Li, M.D. Boston University School of Medicine USA
REVIEW RETURNED	25-Feb-2020

GENERAL COMMENTS	Thank you for giving me the opportunity to review the manuscript entitled, "Protocol for a randomized controlled trial of Subacromial spacers for Tears Affecting Rotator Cuff Tendons: a Randomized, Efficient, Adaptive Clinical Trial in Surgery". This manuscript explains the protocol for a proposed randomized controlled trial with adaptive sample size for more efficient evaluation of the use of arthroscopic debridement with subacromial spacer balloon vs. arthroscopic debridement alone. I have reservations about the utility of this study, since there are many other options that already exist for patients with massive rotator cuff tears, as mentioned by the authors in the introduction. The authors need to justify why these other options may not as advantageous as isolated cuff debridement. I agree with the authors comment in the introduction that rotator cuff debridement is a controversial option. Cuff debridement typically is not a first line of treatment in patients with irreparable rotator cuff repair for a Shoulder & Elbow Fellowship trained surgeon in the US. Rotator cuff debridement is known to be helpful in terms of pain and ADLs, but is not effective in helping patients restore strength or range of motion. Patients undergoing rotator cuff debridement may often require additional surgery. Inspace device may be a helpful alternative to rotator cuff debridement as it is thought to depress the humeral head in patients with cuff tear arthropathy and resist proximal migration of the humeral head. However, using rotator cuff debridement as the comparison group may not be the best comparison to this procedure because it is well known that patients undergoing rotator cuff debridement have very mixed mid to long term outcomes. Also I am concerned for the patients that get randomized to the cuff debridement group as I would suggest for the surgeon to do their best to do a partial repair if possible in
--

	these patients to improve their outcome. Doing a debridement will not do much for outcome. My personal experience is that you can at least fix partially 90%+ of all these patients presenting with massive cuff tears. The authors also note that there is an RCT underway in the US comparing balloon spacer to partial rotator cuff repair. I do not think that conducting another RCT using debridement as a comparator will significantly improve our understanding about balloon spacer effectiveness. I suggest collecting adverse events at an earlier timepoint than at 3 months postoperative. An addition of adverse outcomes reporting earlier in the postoperative period may be helpful. Duration of physical therapy should be documented for every patient. May be helpful to use amount of physical therapy required in each group as part of the cost-effectiveness evaluation. You may want to ask if participants are aware of their allocation at every follow up point as opposed to just at the 12 month timepoint. Can the authors justify why the research MRI for the sub-study will be performed at 8 weeks and 6 months as opposed to at the 1 year mark? *AHD measurements?? Why not standard 20% loss to follow up in sample size determination? All other study methods are sound.
--	--

REVIEWER	pol huijsmans Bergmanclinics Rijswijk The Netherlands occasionally consultant for stryker company
REVIEW RETURNED	01-Mar-2020

GENERAL COMMENTS	Very good and important study. I totally agree with the authors the need for proof of new surgical implants such as the balloon spacer. Some small remarks:  - it is not clear for the reader how and when patients can withdraw from the study - note 38 have been published: Parsons et al. Trials (2019) 20:694 https://doi.org/10.1186/s13063-019-3708-6 - how is maintaining of the correct position of the balloon monitored in this study (I can see this is done for the subgroup with MRI, but other patients could be checked by ultrasound for example, in my experience quite a few of them are already out of position in the early postoperative phase (3 weeks)) -Pre-specified sub-group analyses will be undertaken, but I don't understand the group with a tear <3cm, to me this is not an indication for a balloon spacer, please clarify - which patients are eligible for the MRI sub-group? - why is there no check with X-ray at the final follow-up(acromio-humeral distance, progression of osteoarthritis?) - maybe for the authors interest: at present there is a similar randomized study running in The Netherlands since 2017
--

VERSION 1 – AUTHOR RESPONSE

Reviewer 1

We thank Dr Li for his insightful and detailed review. We feel that some of the comments relate to differences in practice between the United States and Europe, and have tried to clarify these below.

To answer the first point, the balloon has not yet received FDA clearance and so it not yet commonly used in the USA, but is widely used elsewhere in the world. It is an expensive new technology with limited proven efficacy which requires formal evaluation as highlighted by the National Institute for Health and Care Excellence (NICE) in the UK. We therefore politely disagree that the study has poor utility, but understand that in the context of practice in the USA, it is not yet a widely used treatment.

We have added (page 4):

It has not yet received U.S. Food and Drug Administration clearance in the United States of America, but it is a widely used treatment elsewhere in the world.

With regard to partial cuff repair; in this study, if the surgeon finds that the cuff can be repaired (fully or partially), the patient is excluded intra-operatively and not randomised. Therefore these patient have truly irreparable cuff tears, and probably represent an older cohort of patients to those whom partial cuff repair may be an option. Small upper-border tears of sub-scapularis (<1cm) may be repaired. Typically in the UK, younger patients with an irreparable cuff tear would be offered a tendon transfer or superior capsular reconstruction, these younger patients would be excluded from our trial.

Debridement for massive rotator cuff tear remains a recognised treatment in the UK, whereas partial cuff repair is less commonly used, this may be a difference in practice between the UK and US. Partial cuff repairs are occasionally used in the young, but for young patients in the UK reconstructive surgery is more common. Neither debridement or partial cuff repair has a good evidence base, nor is there high-quality evidence to prefer one over the other (Khatri et al, 2019 AJSM, reference 45). After partial cuff repair, pain may improve early on but strength and range of motion often do not, and by 4 years scores decline (Weber SC. J Shoulder Elbow Surg 2017, 26:5 171). Furthermore we feel that debridement is an excellent comparator, as in this trial the balloon is the only differentiating feature between the two arms. If no difference is demonstrated, it can be concluded that the balloon is ineffective.

The USA trial of partial cuff repair compared to the balloon device will give a helpful answer to the reviewer's clinical question but will not resolve the issue of the overall effectiveness of the balloon. Also, it is primarily sponsored by the manufacturer and less applicable to practice where partial cuff tear is a less common procedure. We believe that there is room for an alternative trial and believe that the two sets of results will be complimentary in our understanding of best practice.

Adverse outcomes are reported by sites as soon as they are aware of them, participants are also asked as part of their follow-up as a second method for ensuring events are not missed (for example, treatment at another hospital). We have clarified this by adding (page 8):

as they occur throughout the first 12 months

Physical therapy visits are being collected, and will be included in the health economic analysis as part of healthcare resource use. We have added (page 10):

and the number of physiotherapy visits for each participant will be documented in both arms

We are reluctant to ask participants if they are aware of their allocation until after we have completed the primary outcome to avoid stimulating them to work out which arm they might have been in, also we are not sure this additional information would add value.

The balloon has reliably deflated by 6 months, thank you for spotting this omission, we have added to following line to page 12:

The balloon is expected to have deflated by 6 months by a process of degradation.(59)

We have removed the acronym AHD and replaced with acromio-humeral distance throughout.

We have good rates of follow-up in multiple previous RCTs in our unit, we do not feel that 20% loss-to-follow up should be standard and believe that 15% is fully achievable in this typically older and therefore generally more reliable population.

Reviewer 2

We are very grateful for Dr Huijsman's helpful review, we are delighted to hear that he supports the work.

Thank you for identifying the issue of withdrawals, this was an omission, we have added to page 9:

Participants randomised into the study will be allowed to withdraw from follow-up at any time, without prejudice. This will have no effect on their current or future care.

We have updated reference 38, thank you for spotting this.

If a balloon displaces and this is detected on clinical imaging, we will capture that as an adverse event and report it. We are not monitoring the position of the balloon in all individuals, but are taking a pragmatic approach of assessing clinical outcomes. Normal clinical practice would not include monitoring every balloon for clinical migration with imaging; if patients were asymptomatic then no further intervention would be required. We will report migration when it is identified by clinical teams from their normal practice. We have added the following line:

These include device specific complications such as balloon migration, which will be recorded if it is identified by clinical teams from their normal practice.

We understand the concern about tears of less than 3cm and agree this is likely to be a small group. Whilst most irreparable cuff tears are larger than this, it is possible some may be smaller and these might legitimately be treated with a balloon, or debridement. To ensure the study results can be generalised to all irreparable cuff tears, we have designed the study to capture all of the cases in which a balloon might be used for an irreparable cuff tear. By performing a sub-group analysis, we are answering the reviewers concern, as we will be able to provide data on the balloon in the >3cm group specifically.

Eligibility for MRI sub-study was an omission which we have corrected. We have added (page 12): All participants will be eligible (unless they have a contra-indication to MRI, or do not want to take part in the sub-study), but the sub-study will only be undertaken at a proportion of participating sites.

The accurate measurement of plain radiographs of the shoulder for the purposes of research is known to be very difficult. Acromio-humeral distance is dependent upon the angle of the x-ray especially, and is therefore unlikely to be meaningful in this study. We preferred to include a detailed MRI sub-study that allows a more accurate measurement to be performed on a sub-set of individuals (the costs of doing so on the whole population would be excessive). We have added (page 12):

MRI scans were preferred to radiographs for this purpose as measurements taken from x-rays are prone to error due to variation in the angle between the shoulder joint and the beam of the x-ray.

If a patient has problems and requires further imaging, we will be collecting details of this as an adverse event, or in the hospital resource use information. If a patient is functioning well and clinically requires no further investigation or treatment, we will not be undertaking routine imaging studies.

We have consented participants for longer term follow-up and collection of data from the National Joint Registry and Hospital Episode Statistics. If, at the end of the study, there is an ongoing concern about differential rates of osteoarthritis we will seek funding for longer term follow-up studies.

We are pleased to hear there are other randomised trials on this topic, this will increase the overall knowledge on use of the balloon and best management overall management of irreparable rotator cuff tears. We have not been able to access registration data or protocol details for the study, so have not commented on it in our paper, but are pleased to hear that there are multiple studies on this topic, which will enhance the literature and provide clear advice for patients.